# An Exonic Switch Regulates Differential Accession of microRNAs to the Cd34 Transcript in Atherosclerosis Progression

**DOI:** 10.3390/genes10010070

**Published:** 2019-01-21

**Authors:** Miguel Hueso, Josep M. Cruzado, Joan Torras, Estanis Navarro

**Affiliations:** 1Department of Nephrology, Hospital Universitari Bellvitge and Bellvitge Research Institute (IDIBELL), L’Hospitalet de Llobregat, 08907 Barcelona, Spain; jmcruzado@bellvitgehospital.cat (J.M.C.); 15268jta@comb.cat (J.T.); 2Independent Researcher, Esplugues de Llobregat, 08950 Barcelona, Spain

**Keywords:** atherosclerosis, miR-125b, CD34 cell-clusters, CD34 cryptic splicing, ORF miRNA target, exon switch

## Abstract

Background: CD34^+^ Endothelial Progenitor Cells (EPCs) play an important role in the recovery of injured endothelium and contribute to atherosclerosis (ATH) pathogenesis. Previously we described a potential atherogenic role for miR-125 that we aimed to confirm in this work. Methods: Microarray hybridization, TaqMan Low Density Array (TLDA) cards, qPCR, and immunohistochemistry (IHC) were used to analyze expression of the miRNAs, proteins and transcripts here studied. Results: Here we have demonstrated an increase of resident CD34-positive cells in the aortic tissue of human and mice during ATH progression, as well as the presence of clusters of CD34-positive cells in the intima and adventitia of human ATH aortas. We introduce miR-351, which share the seed sequence with miR-125, as a potential effector of CD34. We show a splicing event at an internal/cryptic splice site at exon 8 of the murine *Cd34* gene (exonic-switch) that would regulate the differential accession of miRNAs (including miR-125) to the coding region or to the 3’UTR of *Cd34*. Conclusions: We introduce new potential mediators of ATH progression (CD34 cell-clusters, miR-351), and propose a new mechanism of miRNA action, linked to a cryptic splicing site in the target-host gene, that would regulate the differential accession of miRNAs to their cognate binding sites.

## 1. Background

Atherosclerosis (ATH) is a chronic inflammatory disease initiated at the vascular endothelium which shows a complex pathogenesis [1]. Endothelium damage displays a focal distribution, particularly at sites of disturbed blood flow, suggesting the contribution of vascular repair mechanisms in ATH progression. Recent studies suggest that endothelial progenitor cells (EPCs) do have an important role in the recovery and repair of the injured endothelium [2], since mobilized EPCs are able to migrate to damaged sites where they would differentiate into mature endothelial cells (ECs) in situ [3]. EPCs come from three major sources; i.e., peripheral blood and bone marrow, the vascular wall, and resident monocytes and macrophages [4,5]. In patients with ATH it has been observed a decrease in the number of circulant EPCs [6], although it is not well know if this is the result from a decreased production or from an increased EPC cell death [7].

Bone marrow-derived EPCs express the progenitor receptor CD34 and are released into the peripheral blood for vascular repair/angiogenesis, since these have reparative potential of endothelial dysfunction [8]. CD34, a membrane glycoprotein, is a critical component of the group of surface receptors that regulate migration and engraftment of progenitor cells to target tissues [9]. *CD34* is expressed as two variant forms after alternative splicing of the so-called “exon-X” at the *CD34* gene [10]. These two forms differ in their cytoplasmic domain, which is almost totally lacking in the variant-1, a fact that abrogates phosphorylation by protein kinase C (PKC) at two sites in the Cterminal cytoplasmic domain [11]. Interestingly, bioinformatic analyses have detected a cluster of miRNA binding sites at the putative 3’UTR of the human and murine *CD34* (Hueso, M., this work) that could be involved in the regulation of the expression balance among both isoforms.

MicroRNAs (miRNAs) are very small RNAs (over 22 nucleotides long) with critical roles in the regulation of gene expression and whose changes in expression have been related to the onset and progression of different diseases (see [12] for a recent review). As of March 2018, the human miRNAome included 1917 mature miRNAs [13]). Aberrant miRNA expression profiles have been described during the progression of ATH, cardiovascular disease (CVD) [14] or renal fibrosis [15]. Thus, plaque progression and rupture were linked to the expression of miR-23a-5p, miR-210 and miR-222 [16,17] while, on the contrary, miR-19b and miR-33a/miR-33b had protective roles on plaque stability [18,19]. Furthermore, plasma miR-144 and miR-33 levels were seen to be increased in coronary artery disease (CAD) patients [20,21], miR-155 was seen to inhibit transformation of macrophages into foam cells by targeting CEH expression [22], and miR-181b was found to be overexpressed in human atherosclerotic plaques and abdominal aortic aneurysms, where it downregulated expression of the tissue inhibitor of MMP-3, and elastin [23]. Lastly, miR-296 has been described as a positive regulator of ATH onset and progression by promoting neovascularization and favoring M1 macrophage polarization [24].

We have previously described the relationship of mir-125b expression with ATH progression. Aortic tissue from ApoE-deficient mice fed with a western diet showed a significant overexpression of miR-125b, but not of the highly related miR-125a. This overexpression was reversed by treating mice with a siRNA against the immune mediator CD40 [25]. MiR-125b belongs to a conserved family (composed by the highly related miR-125a, miR-125b-1 and miR-125b-2) with key roles in cellular differentiation, proliferation and apoptosis [26]. Furthermore, there is experimental evidence of the involvement of the miR-125 family with pathogenesis of ATH. In this sense, it has been described that up-regulation of miR-125a contributed to the differentiation of monocytes towards the inflammatory phenotype [27] through the repression of the TNFAIP3 a negative regulator of NF-kB signaling [28], and that miR-125a was upregulated in oxLDL activated macrophages [29]. Lastly, members of the miR-125 family have been shown to target transcripts whose product could be also involved in mechanisms promoting ATH onset and progression, as STAT3 [30], FOXP3 [31], VEGF [32], MARK1 [33], and BCL2, BCL2L12 and MCL1 [34] among others.

We are interested in the mechanisms of ATH progression, and more specifically in those which involve genes that are implicated in the function of stem or progenitor cells, such as *CD34* or *Lin28a/b*. *CD34* was previously detected as a target of miR-125 in miRNA by bioinformatic analysis (M. Hueso et al., unpublished results) and here we have deepened the molecular relationship among miR-125 and *CD34* in the context of ATH progression. The most significant finding of this work is the description of a new molecular mechanism of miRNA action by which an splicing event at a internal/cryptic splice site of the murine *Cd34* gene would regulate the differential accession of a number of micro-RNAs (including miR-125) to the coding region or to the 3’UTR of the two isoforms of the *Cd34* transcript [10], likely disturbing the expression balance of both CD34 protein isoforms.

## 2. Methods

### 2.1. Ethics Statement

Here, we have used authorized autopsy material from the Pathology Department at the Hospital Universitari de Bellvitge (HUB), L’Hospitalet, Barcelona, Spain. Confidential information from patients was protected following national normatives. This study was performed conforming the declaration of Helsinki, and with the approval by the Clinical Research Ethics Committee of HUB (PR163/13).

### 2.2. Patients and Samples

The characteristics of patients and samples studied in this work have been reported previously [25,35]. Briefly, abdominal aortas were obtained from deceased patients from the HUB, and included atherosclerotic plaques, incipient atherosclerotic plaques, and normal abdominal aortas without injury.

### 2.3. Mice

In this work we used tissue samples from 25 female ApoE^−/−^ mice (Homocigous*Apoe^tm1Unc^* in the C57BL/6 background, Jackson Laboratory, Charles River, Wilmington, MA, USA) that were stored in a previous study [25]. Briefly, 5 mice were considered as controls and sacrificed at 8 weeks (8W). Other 20 mice were i.p. injected with 50 μg of siRNA against CD40 (siCD40), or of a scrambled siRNA (SC) as control, twice weekly. Mice were euthanized by isoflurane anesthesia after 14 or 24 weeks of treatment (SC/14w, *n* = 5; siCD40/14W, *n* = 5; SC/24W, *n* = 5; siCD40/24w, *n* = 5). Mice were fed with a western diet enriched in cholesterol (0.2% *w*/*w*), which provided 42% of calories as fat (TD.88137; Harlan-Tekland, Madison, WI, USA). We isolated the ascendant aortas from 10 ApoE^−/−^ mice treated with the anti-siCD40 siRNA and from 10 ApoE^−/−^ controls, 5 mice treated with the scrambled siRNA (SC) and 5 mice treated with the vehicle (veh) for 24 weeks. This study followed the EU guidelines on animal care and the protocols were approved by the Ethics Committee for Animal Research of the University of Barcelona-HUB. The effect of the anti-CD40 treatment had been previously confirmed by checking the downregulation in the expression of CD40 mRNA and the lowering in the counting of CD40-positive cells (see [25] and references therein).

### 2.4. Reactives

Biotinylated secondary antibodies and vectastain were from Vector (Burlingame, CA, USA). Expression of the isoforms of CD34 (Mn01310773 and Mn00519283) was measured with Taqman Gene Expression Assays (Life technologies, Thermo Fisher Scientific Inc., Waltham, MA, USA). For the anti-CD34 staining, we used an anti-CD34 antibody (RabMab EP373Y; ab81289) from Abcam (Cambridge, UK) following standard procedures (see [25] for details).

### 2.5. Histological and Immunohistochemical CD34 Analysis

Preparation of aortic tissue for protein/RNA expression analysis has been described in detail elsewhere [25,35]. Briefly, aortic sections (aortic arch, descending thoracic aorta, abdominal aorta above renal arteries and abdominal aorta below renal arteries) were split for RNA extraction or for IHC after fixing with 4% PFA. ATH lesions were assessed in sequential, 5 μm thick, sections of the aortic roots. stained with hematoxylin/eosin (H/E) or with a specific antibody by standard procedures. A semiquantitative evaluation of the immunostaining scored the total number of antigen-positive cells with regard of the number of total cells (nuclear H/E staining). Lastly, the ProgResCapturePro 2.7.7 software (Jenoptik, Jena, Germany) was used for morphometric image analysis.

### 2.6. qPCR and miRNA Expression Data

Primary murine miRNA expression data was extracted from the original Taqman Low Density Array card (TLDA) experiment on murine aortas, already described [25], by using the ExpressionSuite Software version 1.1 (Applied Biosystems-Thermo Fisher Scientific, Waltham, MA, USA). RNA extraction from aortas and cDNA conversion have been described previously [25]. Expression of human *CD34* was measured by using Taqman probes in the conditions stated by the manufacturer. The ΔCt/ΔΔCt method was used to compare results with the basal conditions samples (8w mice). For mRNA expression normalization, the housekeeping GAPDH was used, while miRNA expression was normalized by comparison with RNU48 (human) or U6-snRNA (mouse) also from the TLDA data.

### 2.7. Data Mining and Bioinformatic Analysis

The genomic structures of the miR-125a/miR-125b-1 and miR-125b-2 clusters were obtained from the Ensembl genome browser, release 94 [36]. MiRNA sequences were retrieved from the Mirbase [13]. The genomic structure of the murine *Cd34* gene was obtained by “Blasting” the complete murine genome (Annotation release 106) with the cDNA sequences for the two *Cd34* isoforms at the NCBI server. Prediction of Cd34 miRNA binding sites was made at the “Targetscan” for mouse, release 7.2 [37].

### 2.8. Statistics

All data are expressed as mean ±SD. The Kruskal-Wallis test (SPSS 20.0 software, SPSS Inc. Chicago, IL, USA) was used to compare miRNA-expression among basal (8W), control SC-siRNA and anti-CD40 siRNA-treated mice. The Mann-Whitney test was used to compare CD34 expression among siRNA control/treated mice A or in aortas from patients with or without CKD. A value of *p* < 0.05 was considered as statistically significant.

## 3. Results

We are interested in the mechanisms of ATH progression, and more specifically on those that depend on the function of different lineages of stem cells, such as hemopoietic progenitors, or endothelial progenitor cells, characterized by the expression of the CD34 marker. In this context we have recently described an animal model of ATH progression and reversion in which ApoE-deficient mice were fed with a fat-enriched chow to facilitate ATH development and treated for 14 or 24 weeks with an anti-CD40 siRNA to reverse disease progression, or with an irrelevant, scrambled sequence siRNA as control [25]. This disease model was characterized by the upregulation of miR-125b, a member of the miR-125 family, with ATH progression, which was reversed by the treatment with the anti-CD40 specific siRNA [25]. Interestingly, another highly related member of the family, miR-125a, which has been proposed by bioinformatic analysis to target CD34 did not show up in our previous analysis [25]. Here we address the relationship among the members of the miR-125 family and CD34, and describe a new regulatory mechanism by which a cryptic splice event at the CD34 gene could regulate the accessibility of miR-125a (and other miRNAs) to the 3’UTR or to the coding region of the CD34 transcript.

### 3.1. MiR-125: A Complex miRNA Family

miR-125b is encoded by two different genes in human and mice (miR-125b-1 and miR-125b-2), co-clustered with genes encoding members of the miR-99 (miR-99a and miR-99b, miR-100) and let-7 (let-7a-2, let-7c-1 and let-7e) families (see Figure 1A). Furthermore, miR-125b is also highly homologous to miR-125a with which it shares the seed sequence (Figure 1B). Bioinformatic analysis predicted specific promoters for human miR-125b-1/2, but expression of miR-125a seemed to be co-regulated with that of the members of its cluster, miR-99b and let-7e [38]. Given the high homology among miR-125a and miR-125b, and the fact that both shared the same seed sequence but had apparently different roles during ATH progression, we re-evaluated the original data from the TLDA experiment to study changes in miR-125a expression during ATH progression. Figure 1C shows the individual, absolute Ct values obtained for each one of the miR-125a, miR-125b and the housekeeping gene U6-snRNA in the different experimental conditions tested. Furthermore, Table 1 shows the same data expressed as the ΔCt values after normalization to U6 (miR-125-U6). Both clearly show that expression of miR-125a and miR-125b was very similar in the basal, initial condition at 8 weeks (9.43 ± 0.5 cycles vs. 10.47 ± 0.6 cycles; for miR-125a and miR-125b, respectively; *n* = 2), diverged after 24W of ATH progression (6.7 ± 0.6 vs. 3.97 ± 0.6 cycles, respectively; *n* = 3) and returned to almost basal levels after treatment with the anti-CD40 siRNA (7.16 ± 1.6 vs. 6.26 ± −2.0 cycles, respectively; *n* = 3). Given the small differences in the expression of miR-125a and miR-125b, we considered that miR-125a could also have a role, though perhaps less relevant that miR-125b, in ATH progression.

### 3.2. CD34-Positive CellsAre Clustered in Human Aortic Lesions

We had previous evidences of the involvement of CD34 cells in the process of ATH progression (M. Hueso, unpublished results), so that we performed a more complete study of CD34-positive cells during ATH progression. We first analyzed human abdominal aortas from deceased patients by IHQ to detect CD34-positive cells. Figure 2 shows representative views of the human neointimae (Figure 2A,B) and adventitiae (Figure 2D) from aortic sections with ATH plaques, as well as a representative section from a region with no plaques (Figure 2C), all of them stained with the anti-CD34 antibody. It can be clearly seen the presence of clusters of CD34-positive cells in the lesions (Figure 2A,B,D), which at a larger magnification showed a fibroblastoid morphology (arrows at Figure 2B). Interestingly, the non-lesion tissue showed a different distribution of the CD34-positive cells which were non detected in the intima, but scattered in muscle layer (Figure 2C for a representative field). We could not analyze the perivascular adipose tissue (PVAT) because it was lost during preparation of the human samples.

We next dissected human ATH samples into plaque and normal tissue, which were tested by qPCR for mRNA levels of *CD34*. Figure 2E show that *CD34* mRNA levels in the plaque were significantly higher than in normal aortic tissue (ΔCt = 3.5 ± 1.03 cycles in plaque vs. 4.7 ± 0.76 cycles in normal tissue; *p* = 0.05). Since ATH is a frequent complication of Chronic Kidney Disease, we also tested *CD34* mRNA expression in human normal aortas of CDK and non-CKD patients, but only a trend to a lower expression of *CD34* in CKD samples was found (ΔCt = 4.47 ± 0.71 cycles in non-CKD vs. 5.26 ± 0.75 cycles in CKD; *p* = 0.53, see Figure 2F).

We took advantage of the mouse model of ApoE-deficient mice treated with an anti-CD40 siRNA to study the presence of CD34-positive cells in aortic tissues of animals subjected to different treatments. As in humans, we also found clusters of CD34-positive cells in the intima and adventitia of the murine samples (M. Hueso, unpublished result). Furthermore, we had the opportunity of studying the PVAT in these murine samples. As seen in Figure 3A,B individual CD34 cells were found in the perivascular aortic fat (PVAT) of mice treated with the control, scrambled siRNA (Figure 3A) or with the anti-CD40 siRNA (Figure 3B). When different fields were analyzed and counted, it arose that the number of CD34-positive cells was lower in the PVAT of mice treated with the CD40-siRNA when compared with SC-siRNA treated animals (from 14 ± 9% CD34+ cells per field in SC-animals vs. 12 ± 9% CD34+ cells per field in siCD40-treated animals, see Figure 3C), as well as in the adventitia (18 ± 6% CD34+ cells per field vs. 9 ± 3% respectively; *p* < 0.008, see Figure 3C) or in the intima (11 ± 1% CD34+ cells per field vs. 3 ± 6% respectively; *p* < 0.002, see Figure 3C) suggesting that ATH lesions were associated with an increase in the number of tissue resident CD34 cells.

### 3.3. An Exonic Switch Regulates Differential Accession of miR-125, and Other miRNAs, to the CDS or to the 3’UTR of the Cd34 Transcript

We examined the relationship of *Cd34* with miRNAs, among them the miR-125a, during ATH progression. The murine *Cd34* gene is composed by 8 exons and is submitted to a complex splicing regulation. Exon 8 includes an internal/cryptic splice site (I/CSS, Figure 4A) and two in-frame stop-codons (TGA1 and TGA2). Activation of the usual-I/CSS splice sites lead to the expression of two different *Cd34* mRNAs. The transcript variant 1, TV1, links exon 7 to the entire exon 8 and incorporates a premature stop-codon (TGA1 at Figure 4B), which originates a short, truncated form of CD34 lacking almost the entire intracellular domain (Figure 4C). The second *Cd34* transcript, TV2, links exon 7 to the I/CSS (Figure 4A) and incorporates a partial exon 8 which lacks the premature stop-codon and terminates at the TGA2 (Figure 4B) to originate a “full-length” CD34 protein (Figure 4C). In this way, the long transcript (including the entire exon 8) originates a short protein by activating a premature STOP codon, while the short transcript (incorporating a partial exon 8) originates the long CD34 protein (Figure 4B).

Bioinformatic analysis with mouse “Targetscan” of the *Cd34* transcript allowed the detection of binding sites for miR-193, miR-129, miR-125, and miR-351clustered into 70 bp of sequence immediately downstream of the I/CSS. Interestingly, these binding sites would be located at the 3’UTR of *Cd34*-TV1 but at the coding region of *Cd34*-TV2 (Figure 4B and Figure 5A), depending on the splice event that took place, so that a differential splicing would direct the binding of miR-125 (and the other miRNAs) to alternative locations in the *Cd34* transcript, the 3’UTR of *Cd34*-TV1 or the coding region of *Cd34*-TV2 (Figure 5A). Interestingly, two of the miRNAs that targeted the exon 8b of *Cd34*, miR-125 and miR-351 had an almost identical seed region (Figure 5A). Furthermore, another binding site for miR-128-3p was present at the far 3’UTR of both *Cd34* isoforms, but its location was not affected by the splicing event (see Figure 4B).

Analysis of the expression of the *Cd34*-targeting miRNAs (Figure 5B) showed that miR-193, miR-129, miR-125 and miR-128 shared the same expression pattern characterized by an overexpression in SC24W (i.e., disease progression) that was reversed by the anti-CD40 treatment. On the other hand, miR-351 was the only one whose expression decreased in SC24W, with its values not recovering after si-CD40 treatment.

Lastly, we measured expression *Cd34* expression by qPCR in the SC24W and siCD40-24W samples from our disease model, but unfortunately we did not have (basal) control due to sample exhaustion. We could test expression of *Cd34*−TV1 but not of TV2 since there was not a specific commercial Taqman probe for this isoform, so instead we used a probe that amplified both isoforms. The results confirmed our previous results for CD34 mRNA and protein (as number of cells expressing CD34): in the sense that *Cd34*-TV1 and *Cd34*−(TV1 + TV2) were more expressed in disease progression SC24W (TV1: 6.24 ± 0.62 cycles; (TV1 + TV2): 4.73 ± 0.75 cycles) than in the treatment group siCD4/24W (TV1: 7.88 ± 0.37 cycles; (TV1 + TV2): 6.42 ± 0.50 cycles). Furthermore, the difference among *Cd34*−TV1 and *Cd34*−(TV1 + TV2) indicated that both, TV1 and TV2 were expressed.

## 4. Discussion

ATH progression has an hematopoietic component that includes expansion of the hematopoietic stem-cell compartment [39], and we are interested in the role of the stem-cell genes *Cd34* and *Lin28a/b* in the development of atherosclerotic lesions. Originally characterized as an hemopoietic progenitor cell marker [40], CD34 was subsequently detected in vascular endothelial cells [41] and neovascularized tissues [42]. CD34 has been shown to be important in the development of ATH lesions, mainly in neo/re-vascularization events, with CD34-positive cells being detected more frequently in inflammatory-erosive plaques, when compared with plaques of the lipid or degenerative-necrotic plaques [43]. It is thus evident that elucidating the mechanisms of *CD34* expression and CD34-cell differentiation would have a positive impact on CVD disease.

We have previously demonstrated the involvement of miR-125b in the progression of ATH [25], but our data was inconclusive about the role of the other highly related member of the family, miR-125a (Figure 1A,B). It is interesting to notice that *Cd34*, *Lin28a/b* and the miR-125 family shared a molecular link, since data mining of miRNA targets showed that *Cd34* and *Lin28a/b* were predicted targets of miR-125a/b. Furthermore, miR-125a was recently described as a regulator of hemopoietic progenitor cell stemness [44]. In this context, we have addressed the study of the relationship among *Cd34* and miR-125(a,b) with the aim to generate new tools for the study of ATH progression.

We firstly substantiated the involvement of miR-125a in ATH progression by reanalyzing our previous miRNA data to compare expression of the highly related miR-125a and miR-125b. As seen in Figure 1C, while expression of both (in absolute Ct values) was very similar in the control samples, miR-125a was slightly less expressed than miR-125b during ATH progression (SC1-SC3), thus justifying why miR-125a did not show up in our earlier analysis of ATH-progression related miRNA genes. Nevertheless, after treatment with the specific anti-CD40 siRNA both miRNAs showed similar expression levels. The difference in expression levels was confirmed after normalization with the U6-snRNA control (Table 1), and could be likely due to the fact that miR-125b was encoded by two different genes (b-1 and b-2) while miR-125a was encoded by a single gene (Figure 1A).

We next studied CD34 expression in human ATH plaques, as well as in aortic tissues from the mice. CD34-positive cells could be detected as clusters of cells in the neointimas and adventitias of human (Figure 2) and mouse (Figure 3) diseased aortas. Furthermore, individual CD34-positive cells could also be detected in the murine perivascular adipose tissue (PVAT) (Figure 3). Interestingly, in aortic tissue from non diseased aortas the distribution of CD34-positive cells was totally different, being mostly detected as scattered cells in the media layer (Figure 2C). One possibility is that this differential distribution reflected distinct migratory or homing abilities of CD34-positive cells in the context of normal or ATH-diseased aortic tissue.

We next quantified CD34 expression/positive cells by two methods in our two disease models, by PCR (human, Figure 2E) and by directly counting CD34-positive cells (mouse, Figure 3C). Both approaches showed an increase in CD34 in ATH lesions. Increased levels of CD34 mRNA were detected in human ATH plaques when compared with normal aortic tissue (Figure 2E) and more cells were detected in SC24W mice (disease progression) than in siCD40 mice (Figure 3). This is very interesting because most works on CD34-positive PCs and ATH progression has been made with blood-circulating cells and there are scant data on lesion resident cells. Furthermore, a number of authors showed a direct correlation among newly formed blood vessels (as CD34-positive cells) at the neointima and ATH progression [45], that statin treated patients had reduced intraplaque angiogenesis [46], or that a catechin supplementation in the diet of apoE deficient mice resulted in a reduction of ATH lesions and a significant reduction in CD34 expression [47], in line with the results here shown. On the other hand ATH progression has been related with reduced levels of circulating EPCs [48], while a diminution in circulating CD34+/CD45(dim)/VEGFR2- or CD34+/CD45(dim)/CD133+/VEGFR2-cells was shown to highlight patients with coronary endothelial dysfunction [49].

Thus, it is clear that ATH progresses with a simultaneous increase in *CD34* expression and in the number of CD34-positive resident cells, a fact that would be apparently in full contradiction with a role for the miR-125 family in the regulation of *CD34* mRNA stability, since these are also overexpressed in ATH progression. Nevertheless, the end of the coding domain and 3’UTR region of the *CD34* gene is a very complex genomic region, and upon a detailed analysis two features arose that could help to understand the contradictory scenario. Firstly, this region is very poor in predicted miRNA binding sites (after Targetscan), with only five of them, four clustered in a short, 70 bp region. This must be a distinctive feature of the CD34 gene since, on the other hand, only one miRNA (miR-665) has been experimentally confirmed as targeting CD34 [50], although this was not predicted by Targetscan. Furthermore, two of these miRNAs, miR-125 and miR-351, shared the same seed sequence, but only this last showed an expression profile (downregulated in ATH progression) compatible with being a CD34 efector (Figure 5B), thus casting serious doubts on the role of the miR-125 family in the regulation of CD34 stability. It is thus possible that only miR-351, but not miR-125, be involved in CD34 regulation, and that data-mining and bioinformatic analysis were misled by the similarities among their seed sequences. In this sense, miR-351 has been characterized as a developmental regulator that represses differentiation of mesenchymal stem cells (to osteoblasts) by targeting the vitamin D receptor [51], or suppresses angiogenesis [52], roles compatible with its downregulation during ATH progression in our animal model

The second feature to consider when dealing with miRNAs targeting *Cd34* is the structure of its 3’end region. Originally, the *Cd34* gene was shown to be structured in 8 exons [53], and a subsequent work showed that an alternative splicing event at exons X and 8 generated two transcript variants which differed in the Cterminal domain of the protein, generating an isoform almost lacking this domain and another one with a full-length Cterminal domain [10]. This last one included two PKC phosphorylation cytosplamic sites [54], suggesting that the truncated form was unable for signaling. Nevertheless, we have carefully reanalyzed the genomic sequence data of *Cd34* and concluded that the splicing event at exon 8 was more compatible with a splicing at an internal/cryptic splice site than with an alternative splicing as previously described, since the originally defined alternative exon X is in fact part of exon 8 and there is no intronic sequence among them.

This cryptic splicing (exon switch) is intriguing because it would have a compelling consequence on the miRNA binding sites of *Cd34*, since these would be located at the beginning of the 3’UTR sequence at the *Cd34*-TV1, but at the end of the coding region in *Cd34*-TV2 (Figure 4 and Figure 5). This is very interesting because recent work suggested that miRNAs targeting coding regions would cause translational inhibition rather than degradation of the transcript (see [55] and references therein). In this way *Cd34*-TV2 could be also considered as a “miRNA sponge” that would decrease microRNA availability and relieve the repression of target RNAs through sequestering miRNAs away from parental mRNA. Should this be the case, this could have an impact on the balance of CD34 isoform expression, as well as in the function of miR-125s since these could bind to sites in different transcript contexts (3’UTR and CDS) and likely compete with miR-351 for the binding site. This complex miRNA binding region at the 3’ end of the *Cd34* transcript is worth studying “in vitro” with advanced tools of molecular biology to describe its real impact on the dynamics of the *Cd34* transcript and protein as well as on the cells expressing this membrane marker and their role in human disease.

In conclusion, here we have demonstrated an increase in tissue resident CD34-positive cells in the aortic tissue of human and mice during ATH progression. Secondly, although we have failed to provide a sound functional relationship among miR-125 and *Cd34*, we have detected miR-351 as a possible, alternative *Cd34* effector. Lastly, we describe a novel regulatory mechanism of miRNA function by which a cryptic splice event at the *CD34* gene could regulate the accessibility of miR-125a to the 3’UTR or to the coding region of the *CD34* transcript, likely producing different effects on its translation or its stability. Work is in progress to confirm this hypothesis.

## Figures and Tables

**Figure 1 genes-10-00070-f001:**
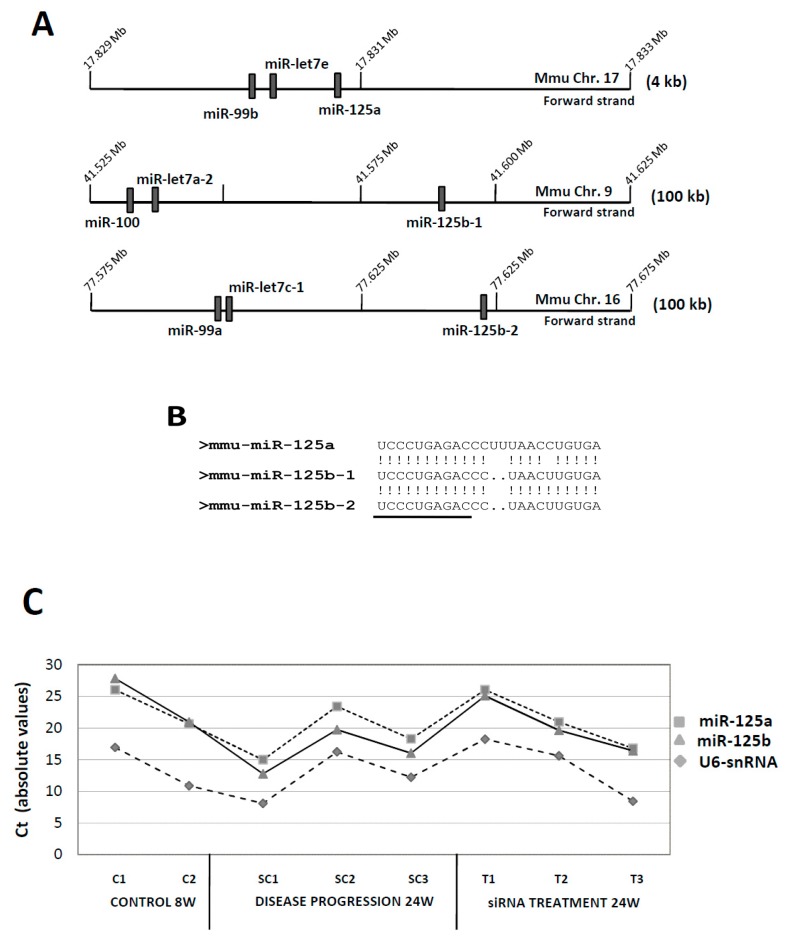
Structure and expression of the murine miR-125a/miR-125b-1/miR-125b-2 gene clusters in ATH progression. (**A**) Graphic diagram of the three miRNA gene clusters. miRNA gene locations (relative positions) were taken from the Ensembl genome browser [36]. Ensemble code for miR-125a was ENSMUSG00000065479, for miR-125b-1 was ENSMUSG00000093354, and for miR-125b-2 was ENSMUSG00000065472. Only miRNA genes and no other coding/non-coding genes are shown. Graphics (genomic regions and gene boxes) are not drawn to scale, and scales are different for each cluster (see numbering for absolute positions in the chromosome). Numbers to the right show the amount of genomic sequence covered by each graph. (**B**) Homology among miR-125a/miR-125b-1/miR-125b-2. Shown are the sequences of the mature miRNAs. (!) stands for a homologous nucleotide, while (.) stands for a deleted nucleotide. Underlined, the seed sequence. (**C**) Absolute expression (without normalization) of murine miR-125a, miR-125b and U6-snRNA (as normalizator) in control, basal samples (8W) in mice treated with the control SC-siRNA (24W, ATH progression) and in mice treated with the specific anti-CD40 siRNA (siRNA treatment, 24W). Shown are the Cts for each individual sample. Expression data was extracted from the original Taqman Low Density Array card (TLDA) experiment [25].

**Figure 2 genes-10-00070-f002:**
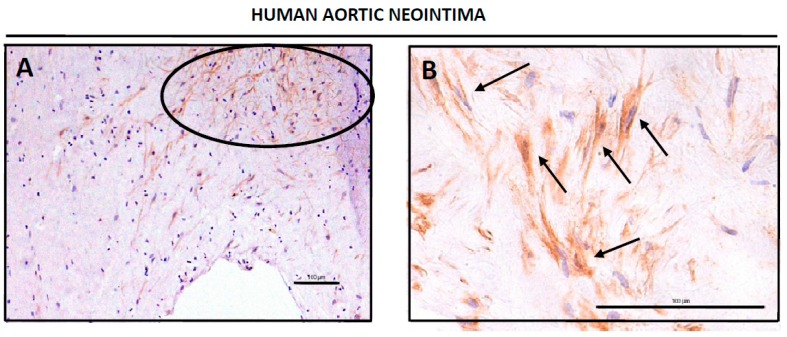
Detection of CD34-positive cells in human aortic tissue. Whole human arteries were isolated, sliced, prepared for IHQ analysis and stained for CD34 as described in Material and Methods. (**A**,**B**) Shown are representative images of two different neointimas, stained for anti-CD34, at different magnifications (20× in 3A, 40× in 3B). The circle in (**A**) shows a cluster of CD34-positive cells, and arrows in (**B**) show±± individual fibroblastoid cells positive for CD34 staining. Tissues were counterstained with hematoxilin-eosin. Bars are 100 μm. (**C**) Representative image of a human aortic tissue from a normal section (non-lesion) showing the intima and the muscular layer (vertical arrows) and stained for anti-CD34. Tissue was counterstained with hematoxilin-eosin. Bar is 100 μm. (**D**) Representative image of a human adventitia stained for anti-CD34. The circle in (**D**) shows a cluster of CD34-positive cells. Tissue was counterstained with hematoxilin-eosin. Bar is 100 μm. (**E**) Expression of *CD34* mRNA measured by qPCR in human ATH plaques compared with normal abdominal aortas. (**F**) Expression of *CD34* mRNA in aortic tissue with normal vascular walls from CKD patients when compared with non-CKD patients.

**Figure 3 genes-10-00070-f003:**
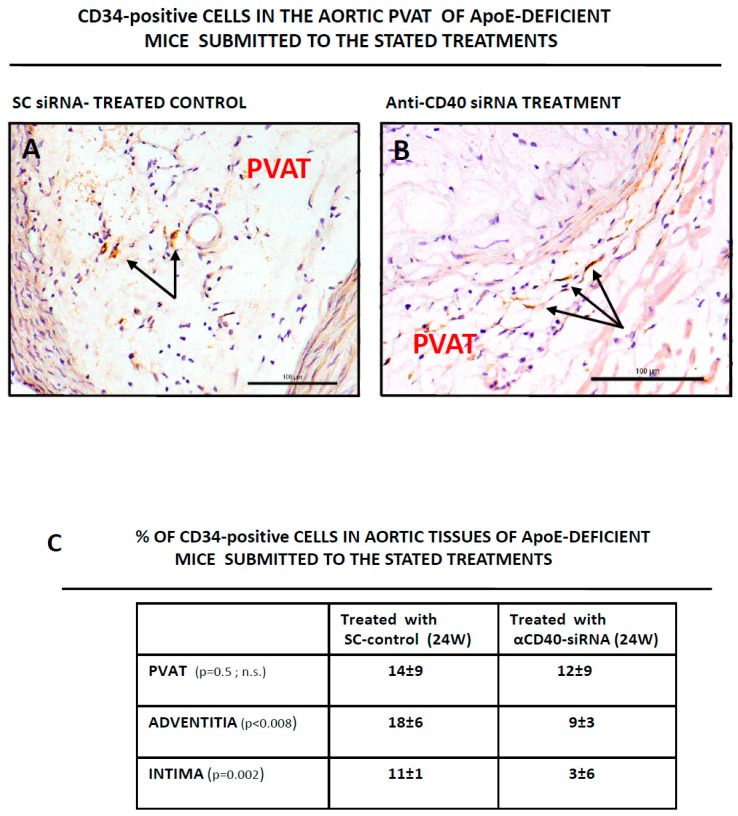
Detection of CD34-positive cells in the murine aortic perivascular tissue. Aortas from ApoE-deficient mice treated with a scrambled (SC) siRNA for 24 weeks as control, or with an anti-CD40 specific siRNA also for 24 weeks, were isolated, sliced, prepared for IHQ analysis and stained for CD34 as described in Material and Methods. (**A**,**B**) Show representative fields with individual CD34-positive cells (arrows) in the SC-siRNA control (**A**) or in the anti-CD40 specific siRNA (**B**). Tissues were counterstained with hematoxilin-eosin. Bars are 100 μm. (**C**) Quantification of CD34-positive cells in the PVAT, intima and adventitia of aortas of ApoE-deficient mice submitted to the stated treatments. A total of 10 fields were examined by two different researchers and the result expressed as % of CD34-positive cells with regard of the total number of eosin-positive nuclei in the field.

**Figure 4 genes-10-00070-f004:**
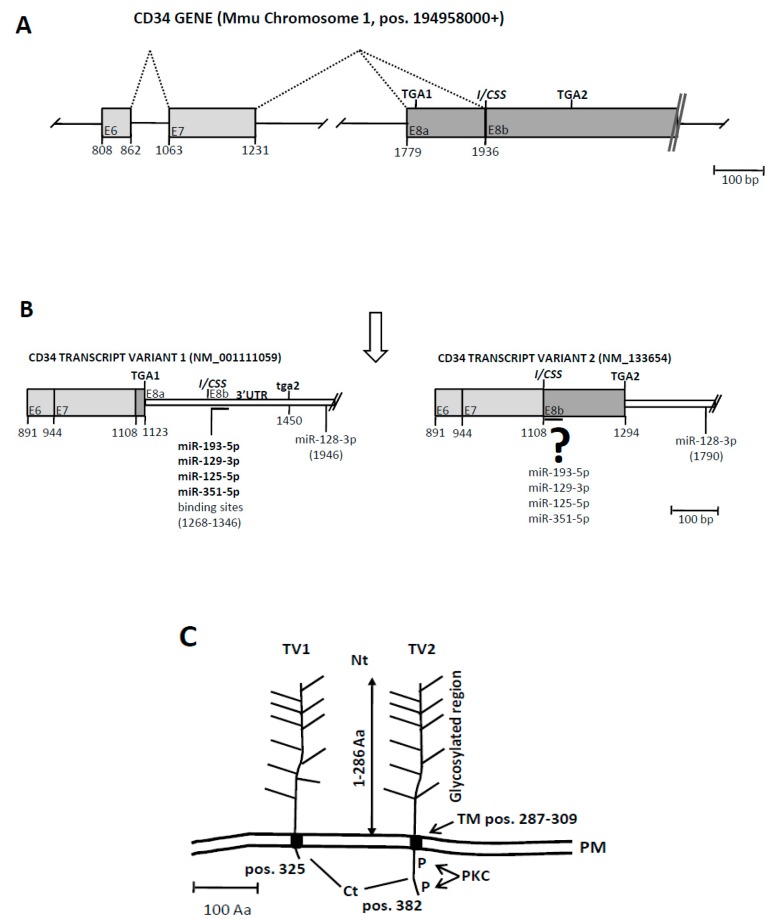
Structure and expression of the murine *Cd34* genetic loci and its interaction with miR-125 and other miRNAs. (**A**) Diagram of the murine *Cd34* gene. Exons are represented by colored boxes, intergenic regions by solid lines and splicing events by doted lines. Shown are exons 6,7 and 8 only. Former exon X is here shown as part “a” of exon 8 (see text). Shown are also the two stop codons (TGA1, TGA2) at exon 8, as well as the internal/cryptic splice site (*I/CSS*) at the virtual juncture of exon 8a/b. Numbering refers to the position in the NCBI Reference Sequence: NC_000067.6. Drawn to scale (1cm = 100bp). (**B**) Diagram showing the two *Cd34* transcripts originated by the internal/cryptic splice event at exon 8. Coding regions are expressed as colored boxes while the 3’UTRs are shown as white bars. Also shown are the two stop codons (the active one in capital letters), the internal/cryptic splice site and the miRNA binding sites predicted by the Targetscanmouse at the 3’UTR of *Cd34* transcript variant 1. Binding sites at the equivalent positions at the coding region of transcript variant 2 are shown under the question mark. Numbering refers to the Genbank accession numbers of the transcripts. Drawn to scale (1cm = 100bp). Ensembl code for *Cd34*-TV1: ENSMUST00000110815. Ensembl code for *Cd34*-TV2: ENSMUST00000016638. (**C**) Diagram showing the two different proteins encoded by CD34 transcripts variant 1 and 2 (TV1, TV2). Shown are the glycosylated Nterminal extracellular domain (Nt), the transmembrane domain (TM) and the different Cterminal domains (Ct) of both protein isoforms. Protein kinase C phosphorylation sites are labelled in TV2. Numbering refers to the Genbank accession numbers of the transcripts (see B). Drawn to scale (1cm = 100Aa). PM stands for plasma membrane.

**Figure 5 genes-10-00070-f005:**
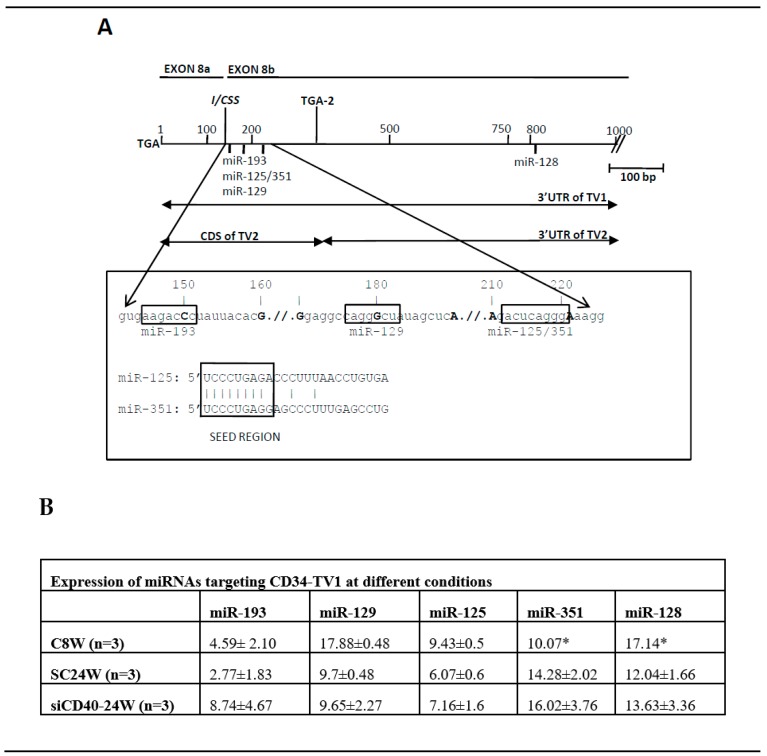
Detailed structure of the *Cd34* region harboring the predicted miRNA binding sites. (**A**) Diagram of 1 kb of sequence at the 3’end of *Cd34*. Shown is the position of the two stop codons (TGA1 and TGA2), the internal/cryptic splice site (I/CSS) and the position of the miRNA binding sites. Expanded is the sequence of the 70 bp, which concentrate the miRNA binding sites. Boxed, the seed sequences of the miRNAs and the comparison among those of miR-125 and miR-351. (**B**) Expression of the miRNAs targeting *Cd34* at different experimental conditions. Data shown after normalization to U6-snRNA. Expression data extracted from the original Taqman Low Density Array card (TLDA) experiment [25]. (*) Only one sample could be analyzed.

**Table 1 genes-10-00070-t001:** Expression of the individual members of the murine miR-125a/miR-125b-1/miR-125b-2 gene clusters in ATH progression and after treatment with the specific anti-CD40 siRNA. The different miRNAs were tested by Taqman Low Density Array card (TLDAs) in the different experimental conditions stated [25], and individual expression values were normalized to U6 expression (ΔCt Gene-U6 snRNA) and expressed as mean ±SD of the different samples. C8W, control mice at 8 weeks. SC24W, mice treated with the scrambled control siRNA for 24 weeks (control group for ATH progression, see Materials and Methods). T24W, mice treated with the anti-CD40 siRNA for 24 weeks (treatment group). (*) Only one sample could be analyzed.

miRNAExpression									
	Cluster 99b/let7e/125a			Cluster 100/let7a-2/125b-1			Cluster 99a/let7c-1/125b-2		
	99b	let7e	125a	100	let7a-2	125b-1	99a	let7c-1	125b-2
**C8W** **(*n* = 2)**	10.95 ± 0.2	8.22 ± 0.2	9.43 ± 0.5	10.64 ± 0.5	13.47 *	10.47 ± 0.6	10.6 ± 0.5	10.8 ± 0.6	10.47 ± 0.6
**SC24W** **(*n* = 3)**	4.51 ± 1.3	3.35 ± 1.8	6.7 ± 0.6	5.57 ± 0.8	7.20 ± 1.8	3.97 ± 0.6	6.32 ± 0.3	3.03 ± 2.2	3.97 ± 0.6
**T24W** **(*n* = 3)**	6.27 ± 2.8	4.99 ± 2.8	7.16 ± 1.6	6.15 ± 1.8	7.5 ± 2.8	6.26 ± 2.0	6.98 ± 2.1	5.74 ± 4.1	6.26 ± 2.0

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
