# Peer review of "An Exonic Switch Regulates Differential Accession of microRNAs to the Cd34 Transcript in Atherosclerosis Progression"

_genes, 2019, doi:10.3390/genes10010070_

Round 1

Reviewer 1 Report

The research article entitled "An exonic switch......Atherosclerosis progression"  shows that CD34 positive cells are increased in the aortic tissue of human and mice during ATH progression". The manuscript is well written but needs some improvement with regard to:

1 Authors need to rewrite methodology section providing adequate  details in each section for readers:

-Why female mice were selected

-Strain of mice selected

-Details of PCR  experiment

2 Are sections included in Figure 2 A, B, C, and D from normal aorta or atherosclerotic plaque area. Please quantify the staining in each section.

3. Authors have mentioned that they used Apo E negative mice. It is recommended to validate same with some experiment. 

4. Authors must show that siRNA for CD-40 has worked. Support findings with experiment

5 There are couple of typographic errors.

Author Response

REVIEWER 1:

1.- Why female mice were selected? Female mice are easier to manage than males, who  are more territorial. Furthermore, there are reports that Atherosclerotic lesions were larger and more advanced in young female than in male ApoE-deficient mice (Caligiuri G et al.,  Atherosclerosis 1999; 145(2):301-8), and this makes genetic analysis easier.

2.- Strain of mice selected. Mice were homocigous Apoetm1Unc  ("Apoe tm1 Unc" ) in the C57BL/6 background, from the Jackson Laboratory, Charles River, Wilmington, MA, USA. This have been added to the main text of the manuscript (lines 98-99).

3. Details of PCR experiment. qPCR were performed with Taqman probes in the standard conditions stated by the manufacturer. This have been added to the main text of the manuscript (lines 132-133).

4.- Sections in Fig 2A,B,C...quantification. We quantified the number of CD34-positive cells but did not include this data because we wanted to highlight the fact that it was not (only) the number of cells but its different distribution (cluster vs. scattered cells) what made the difference in plaque vs. normal tissue. We have included the new panel (Figure 2C) showing the different distribution in the normal tissue to highlight this result.

5.- Validation of ApoE-deficient mice. All mice used in this work were commercial mice from the Jackson Laboratory, Charles River, not breeded  in our animal house. We checked in the microarray experiment that ApoE was not expressed.

6.- Validation that siRNA for CD40 worked. The effect of the anti-CD40 treatment was previously confirmed by checking the lowering of the expression of CD40 mRNA and of the counting of the number of CD40-positive cells. This has been described in our previous articles (Hueso et al.,  Atherosclerosis. 2016; 255:80-89;  Hueso et al., Data Brief. 2016; 9:1105-1112), and added to the main text of the manuscript (lines 110-111).

7.- Typographic errors. We have carefully checked the manuscript and get rid of all typos found.

Reviewer 2 Report

Authors in the manuscript entitled “An Exonic Switch Regulates Differential Accession of microRNAs to the Cd34 Transcript in Atherosclerosis Progression” investigated and extended their previous findings regarding the molecular association between miR-125 and its target gene CD34 in the context of atherosclerotic lesion development. They observed upregulation of CD34+ cells in human and murine aortae during atherogenesis. Furthermore, they identified miR-351 as a new potential target of CD34. Overall, the manuscript is well-written and experiments are well-conducted. However, I have following concerns.

1.    It is not clear to this reviewer - why authors have used only female ApoE knockout mice? Is there any specific reason to use female mice?

2.    Did authors confirmed silencing of CD40 in aortic tissues? If yes, in which layer(s) of aorta, they observed depletion of CD40 expression.

3.    Do expression levels of CD40 and CD34 have any association? It would have been better if authors have performed a co-localization study to determine the expression of CD34 in CD40-silenced cells.

4.    In my opinion, deltaCt is (Ct of gene of interest - Ct of housekeeping gene) (Ct CD34 - Ct GAPDH) and deltaCt is inversely proportional to mRNA expression. Furthermore, y-axis labelling in Fig. 2E and F are not legible.  

Author Response

REVIEWER 2

1.- Female mice. Female mice are easier to manage than males, who  are more territorial. Furthermore, there are reports that Atherosclerotic lesions were larger and more advanced in young female than in male ApoE-deficient mice (Caligiuri G et al.,  Atherosclerosis 1999; 145(2):301-8), and this makes genetic analysis easier.

2.- Silencing of CD40. The effect of the anti-CD40 treatment was previously confirmed by checking the lowering of the expression of CD40 mRNA and of the counting of the number of CD40-positive cells in aortic tissue. This has been described in our previous articles (Hueso et al.,  Atherosclerosis. 2016; 255:80-89;  Hueso et al., Data Brief. 2016; 9:1105-1112), and added to the main text of the manuscript (lines 110-111). Since the siCD40 treatment was systemic and CD40 expression is restricted  to lymphoid cells, it is expected that these were the cells mostly affected by the treatment. Nevertheless, we have not studied in deep the phenotype of the cells affected by CD40-silencing.

3.- Association of CD34 and CD40. This is a very interesting question since there are reports on the regulated expression of CD40 in bone marrow CD34+ hematopoietic progenitor cells (Pyrovolaki K, et al; Arthritis Rheum. 2009; 60(2):543-52), but has not been addressed in this work, mostly dedicated to describe the function of miR-125s during ATH progression. We acknowledge the suggestion that we plan to address in a posterior work.

4.- Delta Ct. We agree with the reviewer. Using negative values was mostly counterintuitive. We have changed all data to positive (gene-GAPDH) values. Figures 2E and 2F have been improved.

Round 2

Reviewer 1 Report

The manuscript can be accepted in present format

Reviewer 2 Report

The authors have addressed the reviewer's comments satisfactorily.